# Prescribing of anti-dementia medications in primary care: A retrospective cohort study in 1489 English General Practices

Charlotte Morris[1,2,3*], Pearl L. H. Mok[2,4], Dame Louise Robinson[5], Louise Allan[4], Darren M. Ashcroft[2,3,6], Tom Blakeman[1,2,3], Evangelos Kontopantelis[7,8]

1 Division of Population Health, Health Services Research and Primary Care School of Health Sciences, The University of Manchester, Manchester, United Kingdom, 2 NIHR Greater Manchester Patient Safety Research Collaboration, The University of Manchester, Manchester, United Kingdom, 3 NIHR School for Primary Care Research, The University of Manchester, Manchester, United Kingdom, 4 University of Exeter Medical School, University of Exeter, Exeter, Devon, United Kingdom, 5 Population Health Science Institute, Faculty of Medical Science, Newcastle University, Newcastle-upon-Tyne, United Kingdom, 6 Division of Pharmacy & Optometry, School of Health Sciences, Faculty of Biology, Medicine and Health, The University of Manchester, Manchester, United Kingdom, 7 Division of informatics, Imaging and Data Sciences, The University of Manchester, Manchester, United Kingdom, 8 Division of Family Medicine, Yong Loo Lin School of Medicine, National University of Singapore, Singapore

* charlotte.morris-3@manchester.ac.uk

## Abstract

### Objective

Evidence suggests that patient-level characteristics such as socio-economic status or ethnicity affect the likelihood of receiving guideline recommended anti-dementia medications. Existing studies often included all-cause dementia, not just the specific subtypes in which medication is indicated. Patterns of prescribing of Acetyl Cholin-esterase Inhibitors (AChEIs) and memantine require further exploration, with little evidence about rates of co-prescribing in English primary care. We examined variations in anti-dementia medication prescribing with patient-level characteristics, and over time.

### Design and setting

Retrospective cohort study, using the Clinical Practice Research Datalink Aurum. Data from 1,489 practices, in England between 2006–2024, were included and linked to patient level Index of Multiple Deprivation data (2019). Cox-regression modelling, clustered at practice level, assessed association between patient-level characteristics and receiving AChEIs, and/or memantine. Time-series analyses examined co-prescribing of memantine and AChEIs.

**Data availability statement:** All relevant codelists are shared within the supplemtary files. The data underlying the results presented in the study are available from CPRD https://www.cprd.com/access-data.

**Funding:** This study was funded by the Wellcome Trust and supported by the National Institute for Health and Care Research (NIHR) School for Primary Care Research (WT6473650) (Charlotte Morris). Tom Blakeman, Pearl Mok and Darren M Ashcroft are funded by the NIHR Greater Manchester Patient Safety Research Collaboration (NIHR204295). The views expressed in this publication are those of the authors and not necessarily those of the NHS, the NIHR, or the Department of Health and Social Care. LA is supported by the Exeter NIHR Biomedical Research Centre and National Institute for Health Research Applied Research Collaboration South West Peninsula. Evangelos Kontopantelis is part-funded by the NIHR HealthTech Research Centre in Emergency and Acute Care (NIHR205301) and the Manchester British Heart Foundation (BHF) Centre for Research Excellence (RE/24/130017). Dame Louise Robinson is affiliated with the NIHR Policy Research Unit in Dementia and Neurodegeneration University of Exeter (DeNPRU Exeter).

**Competing interests:** All authors declare no competing interests.

## Participants

242,007 patients, aged >=18 years, with Alzheimer's or Lewy-Body Dementia, or mixed dementia including one of these subtypes, were included.

## Results

Among the 242,007 patients, 63.1% were prescribed an anti-dementia medication; co-prescribing of memantine and AChEIs peaked at 4.2%. Those in the most deprived quintile were less likely to be prescribed AChEIs (Hazard Ratio (HR) 0.82,0.78-0.86) compared to the most affluent quintile. People with Asian (HR 0.89,0.84−96), or Black (HR 0.79, 0.73-0.86) ethnicities were less likely to be prescribed memantine compared to white people. Those with learning disabilities were substantially less likely to be prescribed AChEIs (HR 0.46,0.42-0.50) or memantine (HR 0.58, 0.50-0.67) compared to those without.

## Conclusion

Overall rates of prescribing of anti-dementia medications were lower than expected. Rates of co-prescription of AChEIs and memantine were low, despite guideline recommendations. We found inequity in anti-dementia medication prescribing, relating to multiple patient-level characteristics highlighting the need for more equitable access to evidence-based treatments.

## Introduction

There are four licensed medications for the treatment of dementia, indicated in Alzheimer's Disease (AD) and Lewy-Body Dementia (LBD). Three are acetyl-cholinesterase inhibitors (AChEIs) (donepezil, rivastigmine and galantamine), alongside memantine an N-methyl-D-aspartate receptor antagonist. Memantine and AChEIs can help manage dementia symptoms, with evidence for their effectiveness demonstrated in randomised clinical trials and systematic reviews [1–3].

UK primary care guidance for anti-dementia medications was first introduced in 2006. It was updated in 2011, and in 2018 [4] (Fig 1). Guidelines recommend AChEIs for mild-moderate AD, and donepezil or rivastigmine in LBD. Since 2006, memantine has been recommended for moderate to severe AD: as monotherapy if AChEIs are unsuitable, and as dual therapy [4]. For LBD, AChEIs are first-line; memantine is considered off-licence if these are not tolerated or contraindicated, for severe LBD, or for potential benefit with specific symptoms [4]. Systematic-review level evidence suggests that the efficacy of AChEIs is actually higher in those with LBD than AD [5], but this may not be well known particularly among non-specialists within primary care, and lead to lower rates of prescribing in this subtype.

While specialists usually commence memantine, primary care teams continue prescribing for stable patients; since 2018 guidance also states primary care clinicians can initiate memantine in addition to AChEIs in eligible patients [4]. To prescribe

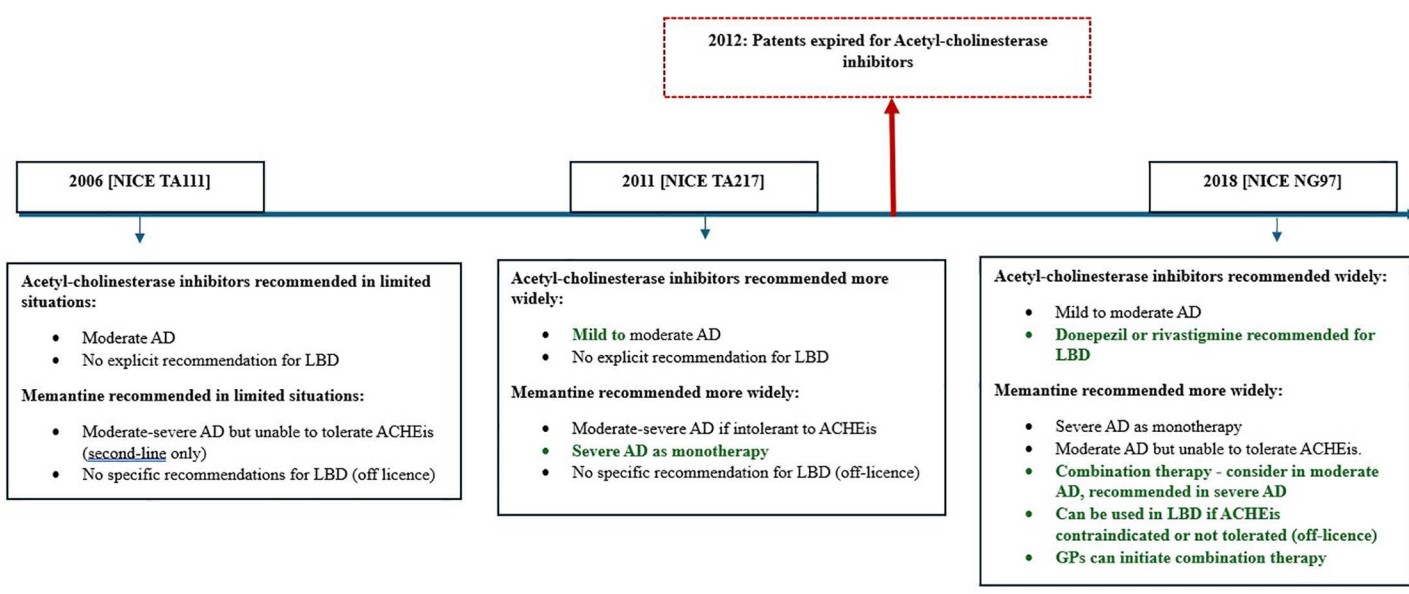

**Fig 1. UK Acetyl-cholinesterase inhibitor and Memantine primary care prescribing guidance over time.**

memantine safely, renal function must be checked, with dose adjustments for impairment. Treatment usually continues long-term if beneficial.

Evidence suggests GPs lack confidence in managing people with dementia [6–8]. There is documented geographical inequity in commissioning of other aspects of dementia healthcare [9], and to what extent this variation is present for anti-dementia medication prescribing, particularly in relation to memantine, requires further inquiry. A 2025 survey found more than 40% GPs were not aware of the guidance for prescribing memantine in primary care [10].

Research has examined patterns of prescribing of AChEIs, and how this varies with patient-level characteristics, such as socio-economic status, ethnicity [11–15]. Many of these papers explored all-cause dementia [16], despite the medications not being indicated in vascular, frontotemporal, and other dementia subtypes. Few papers explored memantine prescribing, usually grouping it in with the AChEIs, or not exploring it at all [15]. This is despite different indications. This study aimed to answer the following questions about anti-dementia medication prescribing in English primary care:

1. How frequently is memantine prescribed in UK primary care for eligible patients, how does this compare with AChEIs?

2. How does prescribing of AChEIs and memantine vary with patient-level socio-economic status, age, sex, ethnicity, diagnostic subtype, and comorbidity profile?

3. How did prescribing patterns of AChEIs and memantine, including switching and co-prescribing, vary over time?

## Methods

### Study design, data source, and linkages

This retrospective cohort study used the Clinical Practice Research Datalink (CPRD) Aurum database of anonymised electronic health records (EHRs) from English primary care practices, recognised for its broad representativeness of the UK population [17]. S1 Fig shows the study design. A patient and public participation group comprising of people impacted by dementia and with an interest in healthcare research were consulted about the study, its design, the outcomes

examined, and dissemination of results. Members of the group met several times, from project design to dissemination. Input was integrated into the final study.

All codelists are provided as appendices. A comprehensive set of dementia codes, developed and compared against a 2024 publication [18] to enhance completeness, was used to identify eligible patients. CPRD Aurum data was linked to patient-level Index of Multiple Deprivation (IMD) 2019 data, and ethnicity data, using a previously published CPRD algorithm [19]. IMD, provided in quintiles to maintain anonymity, is a small area-level deprivation score encompassing seven domains: income, employment, education, health, crime, housing, and environment [20–22]. Individuals aged 18 or over with a dementia code in their EHR between January 1, 2006, and June 30, 2024, who were eligible for patient-level data linkage for Index of Multiple Deprivation (IMD), 2019, and had complete ethnicity data were included. Index-date marked the start of follow-up and date baseline demographics were recorded. It was defined as the patient's first recorded dementia diagnosis after registering with a CPRD-contributing practice, after the study start-date. Patients diagnosed before January 1, 2006, were excluded. Those diagnosed before practice CPRD registration were included, with their index-date set to their practice CPRD registration date.

Follow-up continued until censor-date, defined as earliest of death date, practice registration end-date, or study end-date. Patients whose censor-date preceded their first diagnostic code were excluded. A minimum registration period was not imposed.

We included anyone with a code for AD or LBD at index-date or first diagnosis, including those with multiple codes (classed as 'mixed'). The study protocol received approval from the CPRD Scientific Committee (reference 23_002686). Guidance for reporting studies conducted using routinely collected health data was followed [23].

## Outcomes

Separate codelists for memantine and AChEIs (donepezil, rivastigmine, galantamine) were developed. Those with any memantine prescription recorded after their index-date were identified to create a binary variable, (ever prescribed memantine after the index-date, yes (1) or no (0)). The same was done for the AChEIs.

## Demographics and comorbidities

Comorbidities were defined based on their presence at least one day prior to the index-date (S1 Fig). We assessed comorbidities from the Cambridge Multimorbidity 20 condition score (CMS) [24]. One of these comorbidities is dementia, hence all patients were given this weighting, as having dementia was a pre-requisite to inclusion. Included conditions were anxiety/depression, painful condition, hearing loss, irritable bowel syndrome, asthma, diabetes mellitus, coronary heart disease, chronic kidney disease (CKD), atrial fibrillation, constipation, stroke/transient ischaemic attack, chronic obstructive pulmonary disease, connective tissue disorders, cancer, alcohol problems, heart failure, psychosis or bipolar disorder, and epilepsy. We classed comorbidities as 3 acute conditions (constipation, depression, anxiety), identified if recorded in the year preceding the index-date (days −1 to −365), and 16 chronic conditions. Additionally, learning disability (LD), an important condition not included in the CMS, was identified as a binary indicator using any diagnostic code recorded up to one day before the index-date. As people with renal disease require dose adjustment for memantine, we accounted for CKD separately in the model by also including it as a binary variable. If a condition was not coded, then it was deemed absent not missing.

IMD quintile and ethnicity (Asian, Black, Mixed, White, Other, Unknown), derived from linked CPRD data, were included in models. These represent the best available data for ethnicity within CPRD, and evidence suggests that CPRD ethnicity and IMD data is representative of the overall population and suitable for use in research [19,25]. Geographic data for 9 English Regions (London, North-East, North-West, Yorkshire and Humber, East Midlands, West Midlands, East of England, South-West, South-East (which includes South Central and South-East Coast geographical regions) provided by CPRD was used to classify practice region.

As is customary in electronic health record research, if a condition was not coded, then it was deemed absent not missing. Although there is debate about the accuracy of ethnicity coding within CPRD, we used linked ethnicity date from CPRD itself, derived from their own complex algorithm

### Classifying dementia subtypes

Subtype diagnosis at baseline was used to categorise dementia diagnostic subtype. This was defined as the diagnosis entered at the index-date, or for those with no diagnostic code at the index-date, the first diagnostic code entered.

In the UK, the formal diagnosis of dementia, including subtype, is typically made by a specialist [4]. Therefore, the initial diagnostic code either at the index-date, or their first diagnostic code, is likely to have originated from a specialist letter following a memory clinic assessment. Subsequent codes entered by non-specialist clinicians during routine appointments, where the primary focus might not be on dementia diagnosis, are arguably less precise.

### Identifying subtype diagnoses

To identify subtypes, we divided our comprehensive dementia codelist into separate subtype groups: AD, LBD, Vascular Dementia (VaD), Frontotemporal Dementia (FTD), Unspecified Dementia, Other Dementias. Since patients could have multiple dementia-related codes recorded at different times, we created separate datasets for each subtype. For each patient, we identified their subtype diagnosis as the code entered at index-date, or their first code if no code was entered at index-date.

### Handling Multiple Subtypes

For cases where more than one subtype code was entered on the same day as the index-date or first diagnosis date, we created two additional categories:

- **Mixed (including AD/LBD):** For patients with co-occurring diagnoses of AD and/or LBD with any other type of dementia (e.g., AD and VAD)

- **Mixed (Non-AD/LBD):** For patients with other combinations of dementia subtypes. (e.g., unspecified and FTD)

These mixed categories meant we could focus on those with AD, or LBD in whom anti-dementia medication is indicated. It is important to note while a patient might have multiple codes across different dates, they were assigned to a single index-date (or first diagnosis date) subtype. However, if multiple subtypes were recorded on the same day, they would be classified into one of the "mixed" categories, unless all these codes on the same day referred to a single subtype (e.g., 'AD' and 'AD with hallucinations').

### Assessing Dementia Severity

We were unable to account for dementia severity at diagnosis due to the complexity of coding patterns for dementia within CPRD. There are codes such as 'advanced dementia' or 'mild dementia', however coding practices for even a diagnosis of dementia are variable and suboptimal [26–28]. Evidence is limited, but the accuracy and frequency of severity data is likely even worse [29]. Severity may also be recorded as free text which we were unable to access for analyses. Furthermore, any codes detailing severity may account for severity at diagnosis, but not how the condition progresses so if a mild dementia goes on to become moderate (as is clinically expected), the code may not change. As such we did not assess severity of dementia at baseline, but attempted to account for this within statistical analyses.

### Statistical analysis

All analyses were performed using Stata MP version 18. We conducted time-to-event analyses (Cox regression modelling) to investigate association between patient-level characteristics and likelihood of being prescribed AChEIs or memantine

at least once after index-date. Each outcome was examined in a separate model. Patients were followed from index-date until censor-date. Absence of a code in a patient's record was interpreted as the condition being absent, not missing. We analysed changes over time by examining the number of people ever prescribed AChEIs or memantine by year of first diagnosis, and in each year of the study.

We ran 2 separate models: one examining variation with patient-level characteristics in prescribing AChEIs, one examining memantine. We included only people with AD, LBD, or mixed including AD/LBD dementias. This is because AChEIs and memantine are not indicated in people with other subtypes. We included people with unspecified dementia in sensitivity analyses. A competing risks model was run for both outcomes as a further sensitivity analysis, with death as the competing risk. We ran this on a random 10%-subsample as it was not computationally possible to run for the whole cohort. As is common in large datasets, visual inspection of graphs of covariates over time suggested the proportional hazard assumption was not met. To account for this, we ran further models as sensitivity analyses to support our primary Cox-modelling. These were parametric models, accelerated failure time models, shortened follow-up, and a flexible parametric model with IMD and ethnicity as time-varying covariates.

All models investigating variation with IMD quintile were adjusted for sex, age in index year, modified CMS score at index-date, a binary variable for LD diagnosis, a binary variable for a CKD diagnosis, English region, and ethnicity. Follow-up times were measured in days, with a minimum follow-up of one day. Standard errors were clustered by practice. Data were complete for all variables except ethnicity, and IMD (n = 249 missing; (0.04%)). Given this small number of missing, these cases were excluded and complete case analyses conducted.

Both switching from AChEIs to memantine and co-prescribing both medications are guideline consistent. We examined both patterns using time-series analyses. To examine co-prescribing, we took the first day of every month of the study as a reference date, then established how many people had at least one issue of both memantine and an acetyl-cholinesterase inhibitor in the preceding 28 days. We calculated the number of people in the study on each reference date, then calculated a percentage of people who received both prescriptions in the preceding 28 days.

We then examined the number of people who switched from AChEIs to memantine in each month of the study. Incident switching was operationalised by counting unique patients who initiated memantine (first-ever memantine prescription in the 28-day window prior to the reference date) after having received an acetyl-cholinesterase inhibitor in the preceding three months, while excluding any prior memantine use. The prevalence of those who switched was operationalized by counting the number of unique patients who, for each given month, were actively prescribed memantine (prescribed in the preceding 28 days) and not co-prescribed AChEIs (in the same 28 days), provided they had a documented history of having used an acetyl-cholinesterase inhibitor at any time prior to the current 28-day period. This was also examined for people switching from memantine to AChEIs. S2 Fig details how each indicator of prescribing was operationalised.

## Results

242,007 people were eligible for the study after applying inclusion/exclusion criteria. The derivation of the cohort is shown in Fig 2. Table 1 shows the cohort's demographic profile, and the number prescribed AChEIs, memantine, and both medications by patient-level characteristic. In total, 152,647/242,007 (63.1%) people were ever prescribed an acetyl-cholinesterase inhibitor or memantine.

### Acetyl-cholinesterase inhibitor prescribing

Overall, 114,498/242,007 (47.3%) were ever prescribed an acetyl-cholinesterase inhibitor. By subtype this was: 96,212/204,188 (47.1%) with AD, 10,887/18,384 (59.2%) people with LBD, and 7,399/19,435 (38.1%) with mixed AD/LBD. There was inequity in the likelihood of receiving a prescription for an acetyl-cholinesterase inhibitor (Fig 3). Compared to those in the least deprived IMD quintile, those in the most deprived were 18% less likely to be prescribed an acetyl-cholinesterase inhibitor (HR 0.82, CI 0.78-0.86). People of Asian (HR 0.88, CI 0.84-0.93), Black (HR 0.90, CI 0.85-0.95),

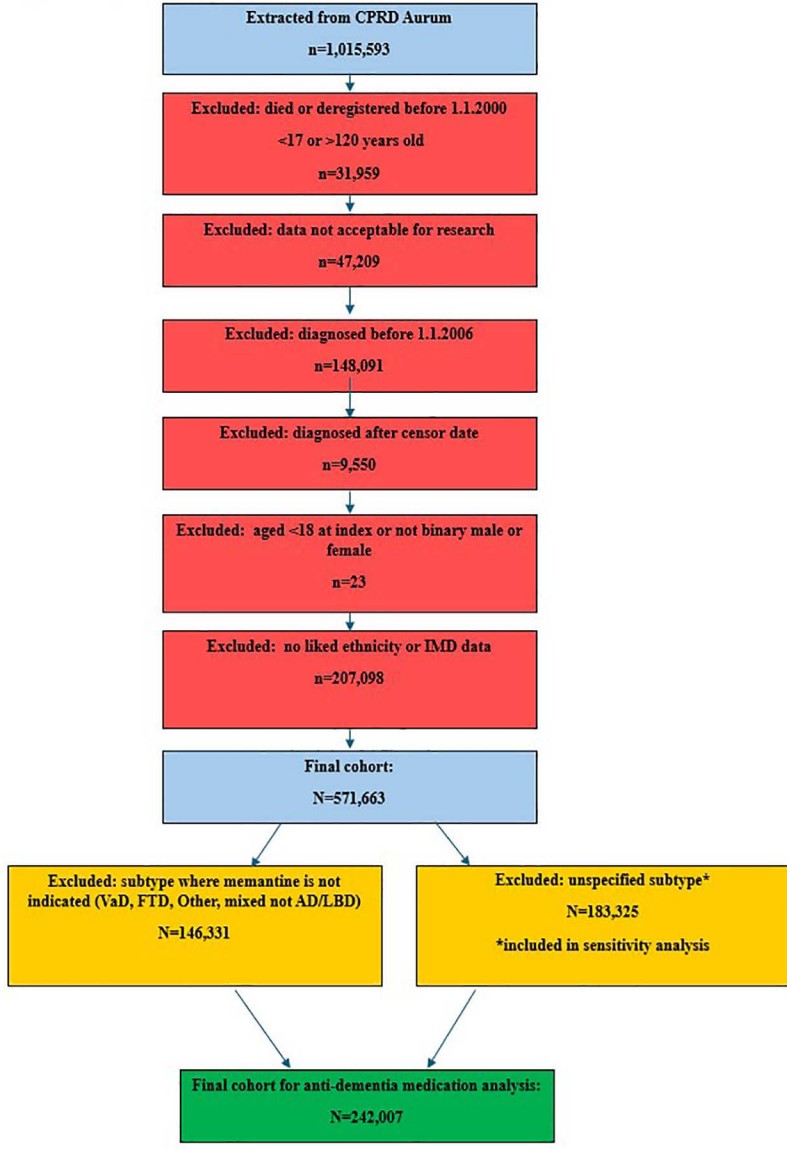

**Fig 2. Study flow diagram.**

and unknown ethnicity (HR 0.84, CI 0.78-0.91), were all significantly less likely to be prescribed AChEIs than white people. Those with LD were 54% less likely to be prescribed AChEIs (HR 0.46, CI 0.42-0.50) compared to those without. Regional differences were seen, with those in the West Midlands (HR 0.86, CI 0.79-0.93) and East of England (HR 0.83, CI 0.75-0.92) being significantly less likely to prescribe the medications, compared to London. Those in the North East (HR 1.26, CI 1.14-1.39) were significantly more likely. Small differences were seen with CKD, age in index year, comorbidity profile, and sex.

## Memantine prescribing

Overall, 58,322/242,007 (24.1%) ever received a memantine prescription; of these 38,149/242,007(15.8%) were prescribed memantine alone as a first-line treatment without receiving a prescription for an acetyl-cholinesterase inhibitor.

**Table 1. Baseline characteristics of the sample.**

| | Total | N receiving memantine once or more (%) | N receiving AChEIs once or more (%) |
|---|---|---|---|
| **Total** | 242,007 | 58,322 (24.1%) | 114,498 (47.3%) |
| **Female** | 151,901 | 34,466 (22.7%) | 71,267 (46.9%) |
| **Male** | 90,106 | 23,856 (26.5%) | 42,231 (48.0%) |
| **IMD Quintile (n, %)** | | | |
| 1 (least deprived) | 55,912 | 13,497 (23.1%) | 28,059 (50.2%) |
| 2 | 55,641 | 13,454 (23.1%) | 26,709 (48.0%) |
| 3 | 47,582 | 11,335 (19.4%) | 22,236 (46.7%) |
| 4 | 43,076 | 10,320 (17.7%) | 19,665 (45.7%) |
| 5 (most deprived) | 39,796 | 9,716 (16.7%) | 17,829 (44.8%) |
| **Ethnicity** | | | |
| Asian | 4,968 | 1,204 (24.2%) | 2,313 (46.6%) |
| Black | 4,614 | 977 (21.2%) | 2,107 (45.7%) |
| Mixed | 711 | 154 (21.7%) | 348 (49.0%) |
| White | 229,155 | 55,449 (24.2%) | 108,655(47.42%) |
| Other | 144 | 37 (25.7%) | 61 (42.4%) |
| Unknown | 2415 | 501 (20.8%) | 1,104 (42.0%) |
| **LD** | | | |
| Yes | 1,297 | 258 (19.9%) | 554 (42.7%) |
| No | 240,710 | 58,064 (24.1%) | 113,944 (47.3%) |
| **CKD diagnosis** | | | |
| Yes | 45,205 | 9,968 (22.1%) | 20,439 (45.2%) |
| No | 196,802 | 48,354 (24.6%) | 94,059 (47.8%) |
| **Subtype** | | | |
| AD | 204,188 | 50,805 (24.9%) | 96,212 (47.15%) |
| LBD | 18,384 | 2,361 (12.8%) | 10,887 (59.2%) |
| Mixed (including AD and/or LBD) | 19,435 | 5,156 (26.5%) | 4,362 (22.0%) |
| **English Region** | | | |
| London | 28,400 | 6,981 (24.6%) | 13,370 (47.1%) |
| North East | 11,562 | 3,314 (28.7%) | 6,410 (55.4%) |
| North West | 53,510 | 14,973 (28.0%) | 24,912 (46.6%) |
| Yorkshire and Humber | 9,107 | 2,191 (24.1%) | 4,529 (49.7%) |
| East Midlands | 4,211 | 819 (19.5%) | 2,006 (47.6%) |
| West Midlands | 36,073 | 7,035 (19.5%) | 15,626 (43.3%) |
| East of England | 10,599 | 1,634 (15.4%) | 4,662 (44.0%) |
| South West | 32,596 | 7,756 (23.8%) | 15,549 (47.7%) |
| South East | 55,949 | 13,619 (24.3%) | 27,434 (49.0%) |
| **Mean age at diagnosis [median, range]** | [83, 19-107] | 81 [82, 19-104] | 80 [82, 28-107] |
| **Mean co-morbidity score [median, range]** | 5.2 [1.8-33.5] | 5.2 [3.6, 1.8-33.5] | 5.1 [3.8, 1.8-29.8] |
| **Median follow-up time (days), [mean, range]** | 672 [902,1-6753] | 842 [1036, 1-6611] | 890 [1101, 1-6753] |
| **Median time to event [mean, 25%, 75%, range]** | – | 58 [309, 11-344, 1-6559] | 30 [165, 8-131,1-5856] |

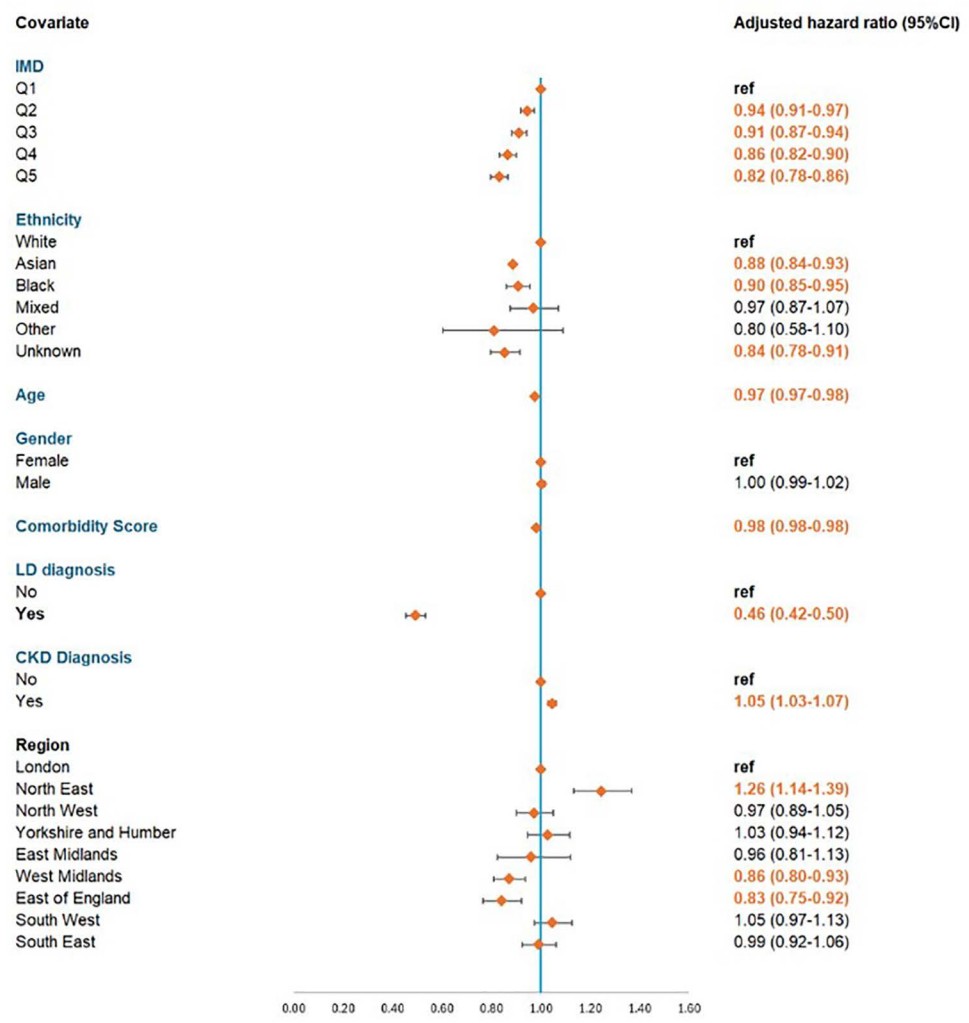

Footnote: Quintile 1 is least deprived, quintile 5 is most deprived.

**Fig 3. Adjusted hazard ratio for likelihood of ever receiving an acetyl-cholinesterase inhibitor based on patient characteristics (n = 242,007).**

By subtype this was: 50,805/204,188 (24.9%) people with AD; 2,361/18,384 (12.8%) people with LBD, and 5,156/19,435 (26.5%) people with mixed AD or LBD. No clinically significant differences were seen with IMD, comorbidity score or age. Those with Asian (HR 0.89, 95% CI 0.84-0.96), Black (0.79, 0.73-0.86) or Mixed (0.85, 0.73-1.00) ethnicity were significantly less likely to be prescribed memantine than people with white ethnicity. Males were significantly more likely to be prescribed memantine than females (1.22, 1.19-1.24), and those with a LD were significantly less likely to be prescribed memantine (0.58. 0.50-0.67). Those with CKD were less likely to be prescribed memantine (0.91, 0.88-0.93). Significant differences were seen by region, compared to those in London: those in the North-East (HR 1.19, CI 1.05-1.34), and North-West (HR 1.17, CI 1.06-1.28) were more likely to be prescribed memantine. Those in the East Midlands (HR 0.72,

CI 0.63-0.83), West Midlands (HR 0.78, CI 0.70-0.88), East of England (HR 0.58, CI 0.49-0.69) were significantly less likely to be prescribed memantine (Fig 4).

## Changes in prescribing over time

Rates of prescribing by year of first dementia diagnosis were examined for both outcomes (S1 Table). For AChEIs, there was a notable peak for those diagnosed in 2013 before decreasing again until 2020. In contrast, memantine prescribing increased gradually between 2006–2021, before decreasing from 2021 onwards.

We also examined the number of people initiated on memantine or AChEIs in each year of the study (S1 Table). For memantine, rates of initiation increased until 2021 before dropping off. For AChEIs, due to small overall numbers in the

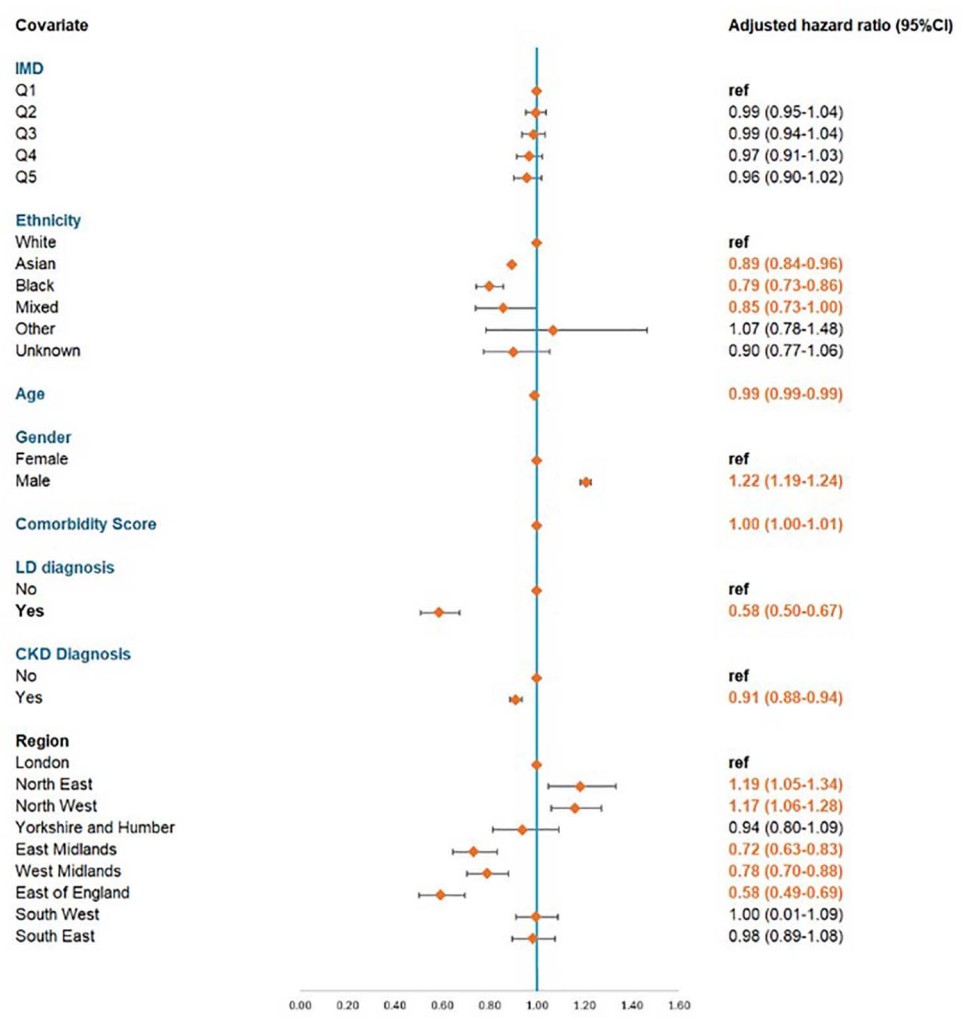

Footnote: Quintile 1 is least deprived, quintile 5 is most deprived.

**Fig 4. Adjusted hazard ratios for ever receiving memantine based on patient characteristics (n=242, 007).**

early years of the study, the percentage of initiations was highest in 2006. This ranged from 17.2%−19.8% between 2008–2014, before gradually dropping off from 2014 onwards.

## Switching and co-prescribing of anti-dementia medication

The time-series showed the percentage of people co-prescribed memantine and AChEIs increased over time (Fig 5, S2 Table). Prescribing of both medications peaked in the 28 days preceding 1/2/2022, when 4.2% patients received at least one issue of both prescriptions. A noticeable acceleration occurred in early 2011, loosely coinciding with guideline changes. The percentage of people receiving both prescriptions rose from 2% to just over 4% between 2018–2023 again, broadly coinciding with guidance published in 2018.

The analysis of incident switching from AChEIs to memantine (Fig 6, S2 Table) showed switching was uncommon early in the study, with an increase starting around 2011. After 2015, the incidence of switching stabilised at around 0.06%–0.07% though with high month-to-month variation. The data for prevalence of switching (Fig 7, S2 Table) show rates below 0.5% until 2012, but then with a period of sustained growth until 2018. The number switching from memantine to AChEIs peaked at <5 patients per month; numbers too small to conduct any further analysis.

## Sensitivity analyses

When we included those with an unspecified dementia diagnosis as a sensitivity analysis, 154,860/425,332 (36.4%) were prescribed an acetyl-cholinesterase inhibitor, 79,304/425,332 (18.6%) people were prescribed memantine. Both models were run again, including people with unspecified dementia. Overall, the results were very similar in terms of direction and magnitude, to the main analyses (S3 Fig).

Prescribing at the practice level was examined for AChEIs, memantine, and both. S4a-S4b Fig show violin plots illustrating the proportion of patients in each practice prescribed AChEIs, and memantine, and how this varied by patient-level deprivation quintile. There was variation in prescribing rates noted (S4 Fig), which was accounted for by clustering standard errors at the practice level in the models.

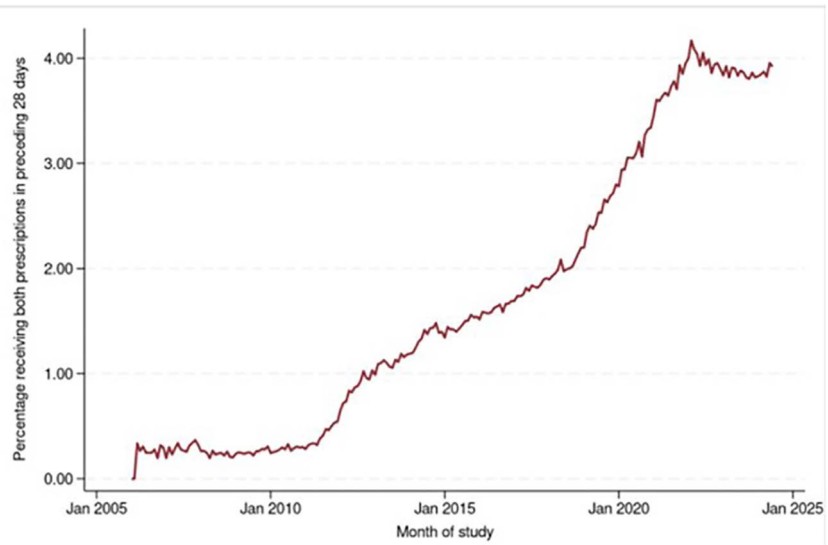

**Fig 5. Time-series graph showing the number of people receiving both an ACHE-inhibitor and memantine in the preceding 28 days, for every month in the study (Jan 2006-June 2024).**

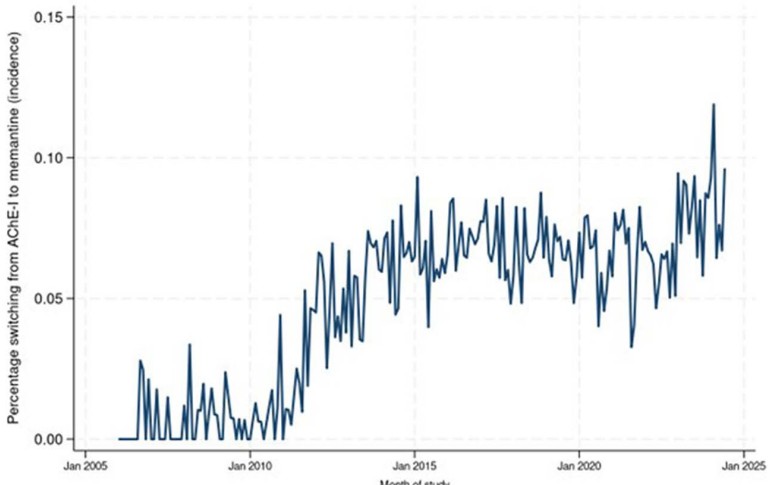

**Fig 6. Time series graph showing percentage of people switching from ACHE-I to memantine in each month of the study (incidence of switching).**

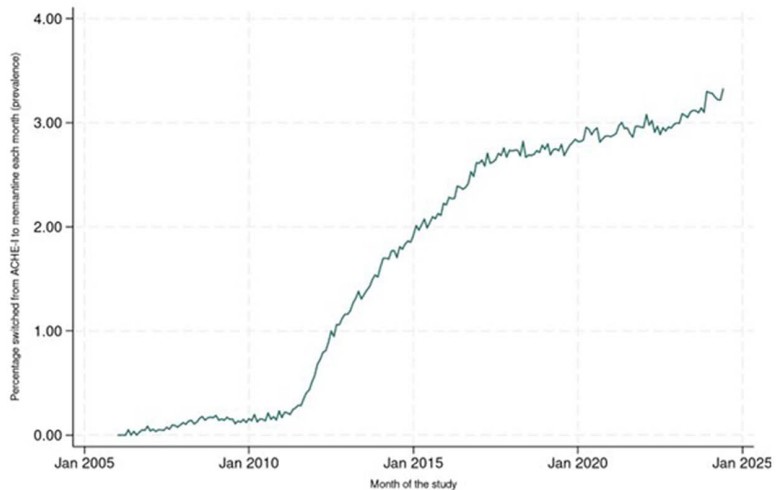

**Fig 7. Time series graph showing percentage of people who have switched from ACHE-I to memantine in each month of the study (prevalence).**

Competing risks models were run with death as the competing risk, for both outcomes (S5 Fig). The relationship with IMD remained strong for AChEIs. The relationship was also significant for memantine with those in the most deprived quintile being significantly less likely to be prescribed memantine compared to those in the least deprived (HR 0.70, 0.59-0.82), in contrast to the primary model where the relationship was not significant. The associations for LD remained strong for both outcomes, as did the findings for regional differences. The relationship for ethnicity was attenuated for both outcomes, perhaps because of the small sample size, especially in the non-white ethnicity subgroups. As the competing risks models were run on only a random 10% of the sample, the results should be interpreted cautiously, but do support our primary Cox modelling.

As we used index-date (where the patient had a diagnosis of dementia and was registered with CPRD) some people's date of entering the study was after their first dementia diagnosis date. Some of these people were diagnosed with dementia at their original, non-CPRD practice, started on an anti-dementia medication, and then this stopped and never restarted when they entered the study on their CPRD registration date. We examined these cases further. Considering those with subtypes where anti-dementia medication is indicated, 18,164 people entered the study on their CPRD registration date (7.51%). The numbers prescribed anti-dementia meds before their index-date and never again after were very small. The number of people prescribed AChEIs changed from 114,498 (47.3%) to 116,097 (47.9%). The number of people prescribed memantine changed from 58,322 (24.2%) to 58,821 (24.3%). Overall, these differences were deemed too small to affect the results. Furthermore, as these drugs were prescribed before practice started contributing research standard data to CPRD these should be interpreted with caution.

We visually examined graphs of covariates to examine if the proportional hazards (PH) assumption was violated for each of the three models. As expected, given our very large dataset with long follow up, visual inspection confirmed violation of the assumption for covariates, mainly at longer follow-up times. To ensure the robustness of the interpretations from the Cox-model for this dataset, we completed detailed sensitivity analyses.

Firstly, we used a parametric regression model, (Weibull distribution). This gave very similar results for both examined outcomes (S6 Fig) with no changes in significance or the direction of the relationship seen for the covariates of interest. We also examined accelerated failure time models (lognormal distributions) for each of the outcomes. These showed that those in lower IMD quintiles (compared to the highest IMD quintile), with an LD diagnosis (compared to those without), and with Asian or Black ethnicity (compared to white) had significantly longer times to prescribing the medications, supporting the finding of the Cox models, and suggesting access to these medications for people with these characteristics was more difficult (S3 Table). We ran the analyses with a shortened follow up of 24 months after the index-date, which again showed very similar results in terms of magnitude, significance and direction (S7 Fig). Finally, a flexible parametric model was run [30,31], with IMD and ethnicity included as time-varying covariates. In the non-time varying covariates, direction and significance of the relationships supported the findings of the Cox models robustly. For IMD and ethnicity, the time-interaction co-efficient supported the findings of the Cox modelling, with the effects of deprivation and minority ethnicity appearing to worsen over time (S4 Table). Our multiple, detailed sensitivity analyses support the robustness of our primary Cox models for the dataset, despite the PH assumption violations.

## Discussion

### Summary of results

Prescribing rates were lower than expected, with 63.1% people receiving anti-dementia medication. We observed important disparities with patient-level characteristics. Individuals from the most socioeconomically deprived areas were less likely to be prescribed AChEIs. People of Asian, Black, and Mixed ethnicities, and people with LD, were less likely to be prescribed AChEIs or memantine. Findings highlighted regional variations and sex disparity. Overall, our study demonstrates substantial inequity in access to treatment linked to patient-level characteristics.

Despite co-prescription of AChEIs and memantine being recommended in severe AD, and that it should be considered in moderate AD, very small numbers received this (4.2%). This is despite guidance memantine can be started as cotherapy in primary care without specialist input.

### Results in the context of other literature

We found socio-economic status, measured with IMD, was strongly associated with the likelihood of prescribing AChEIs, but not with the likelihood of prescribing memantine, contrasting with previous studies [32,33]. This may be due to our inclusion of people with LBD as well as AD, or our use of IMD as the marker of socio-economic status both different methodologies to existing studies. Most evidence suggests deprivation is linked to a reduced chance of receiving anti-dementia

medications overall, but this finding is not ubiquitous [15]. Our findings support the importance of considering subtype diagnosis and different anti-dementia medications separately when exploring prescribing patterns [15]. This is evidenced by our sensitivity analysis including people with unspecified dementia, where those in the most deprived quintiles were significantly less likely to be prescribed memantine. When we considered only subtypes where memantine is indicated the relationship was not significant. It is unclear why inequity with IMD was seen for AChEIs, but not with memantine alone. We hypothesize it may be due to people in poorer areas having earlier mortality before their dementia becomes severe enough to require memantine, a theory supported by our competing risks model.

The finding of disparity of prescribing AChEIs and memantine with ethnicity is in keeping with other work, mainly examining anti-dementia medication prescribing generally (rather than memantine separately), which found people with non-white ethnicity were significantly less likely to receive anti-dementia medications [34,35]. It is unclear whether these variations are due to differences in dementia severity, or more socially driven – due to bias, healthcare-seeking behaviours, resourcing, or access issues.

People with LD being 42% less likely to ever be prescribed memantine, and 54% less likely to be prescribed an AChEI suggests gross disparity. Although inequalities in healthcare for people with LD are described [36], this is an important and novel finding relating to anti-dementia medications. As many people with LD develop dementia (particularly AD), this is important. The reduced rate of prescribing memantine in those with CKD is expected given prescribing guidance.

Evidence suggests prescribing of anti-dementia medications has increased for those diagnosed in recent years but rates are still below what we would expect [37]. This is in keeping with our findings. NICE guidance suggests people with moderate to severe AD should be on memantine, we would expect 2/3 of people with AD to meet criteria for this. Given everyone in our study was eligible to be offered one of these medications based on diagnostic subtype, the low prescribing rate is concerning. It seems unlikely that such a high percentage declined or did not tolerate either medication, or both medications were contraindicated.

Regional differences may be related to differences in commissioning [9] or driven by clustering of other intersectional factors associated with health disadvantage [38] such as low socio-economic status and minority ethnicities in inner-city areas.

Rates of prescribing of AChEIs fluctuated over time, with peaks seemingly corresponding to guidance changes (e.g., for those diagnosed in 2007−8, after 2006 guidance, and those diagnosed in 2011−13 after 2011 guidance). Rates of memantine prescribing declined for those diagnosed between 2022−24 compared to 2021. This is likely because memantine is typically second-line for moderate-to-severe dementia; those with shorter follow-up may not have progressed to that stage. Interestingly, 17.8% of people diagnosed in 2024 were prescribed memantine, meaning their maximum time to initiation of memantine was 6 months (until study end-date). This could suggest that the drug is being used earlier, or that dementia diagnoses are being made late when the condition is already advanced.

The study covers the period of the Covid-19 pandemic (2020−22), however the time series analyses did not show any dramatic changes in prescribing levels over these months. This may be because prescribing continued, or because there were fewer diagnoses and fewer prescriptions issued, meaning overall percentages remained the same. Other large important studies going on during this period included NHS England drives to diagnose dementia [39,40] and changes to guidance as shown in Fig 1 Despite this we did not see notable shifts in monthly prescribing levels that corresponded with these events.

We are unaware of previous work examining co-prescribing of memantine and AChEIs over time; the low rates seen are concerning. By no means are all people eligible for co-prescriptions receiving these based on our findings. For switching from AChEIs to memantine, numbers were lower than expected. The most sharp and sustained increases seen in monthly rates for all three measures broadly corresponded with guideline changes.

## Strengths and limitations

The key strengths of this study were the large cohort and use of diagnostic subtype ensuring anti-dementia medications were indicated. We used Cox regression to adjust for key variables and time. Analysing even a single anti-dementia

medication prescription was a strength as it captures the initial prescribing decision. Separating memantine to explore prescribing patterns of this medication in addition to AChEIs is novel within this dataset. A further strength was our investigating co-prescribing rates of memantine and AChEIs, which to our knowledge has not been previously examined. Our detailed sensitivity analyses (parametric regression modelling, accelerated failure time modelling, shortened follow-up time, and flexible parametric models) all supported the results of our primary Cox regression models, suggesting robustness of the findings.

The study has limitations. Dementia severity could not be determined due to complex coding in CPRD. This meant we could not determine eligibility for medications at baseline, or whether AChEIs or memantine should be offered. However, because dementia is a progressive condition a single baseline severity measure may not reflect eligibility for treatment over the entire study period. In clinical reality, most people with a diagnosis would be eligible for at least one of the examined medications at some point during the clinical trajectory (AChEI if diagnosed with mild or moderate; memantine if diagnosed when severe). This is an important limitation and highlights the need for more detailed, standardised dementia coding within primary care. Our reliance on baseline diagnostic codes is a limitation; it may have excluded cases who later received a subtype diagnosis where anti-dementia medication was indicated; the sensitivity analysis including those with 'unspecified dementia' at baseline helped mitigate this. We took absence of a diagnostic code as absence of a condition in terms of the comorbidity score. Although this means we may have underestimated the comorbidity burden for some patients, as we had very wide ranging codelists and we used a single instance of the code as identifying the comorbidity, the number of patients affected is likely to be minimal.

We were unable to determine if memantine prescriptions were initiated in primary care or by specialists, an important distinction given low awareness among GPs [10]. Generally, in the UK, AChEIs and memantine are initiated in secondary care, with continued prescribing in primary care. We did not examine rates of referral or initiation in secondary care. Therefore, inequities seen may reflect disparities in access to secondary care, rather than primary care prescribing. Despite this, primary care clinicians are responsible for care planning for people with dementia and should regularly be ensuring they are on the right medications. Using index-date rather than diagnosis date meant a small number of cases who were prescribed anti-dementia medications before they entered the study and never again after, were classified as never having a medication. However, on exploration, differences in prescribing were very small (<0.6%) and deemed unlikely to influence results. The small number of patients with LD in our cohort, likely due to their lower life expectancy and the older average age of our cohort, means these findings should be interpreted as a strong signal for further research. We did not assess if patients were offered AChEIs or memantine and declined or did not tolerate the medications. This was due to the complexity, accuracy and variability of how this was coded limiting inference.

### Implications for practice, policy and research

Low prescribing rates for licensed, guideline-recommended dementia medications are concerning. This suggests many people with dementia are not receiving guideline-recommended treatments. This was particularly true for co-prescribing of memantine and AChEIs. This is compounded for key subgroups, such as those in socio-economically deprived areas, with minority ethnicity, and LD. New disease modifying anti-amyloid drugs [41,42] are currently unavailable on the NHS, due to a lack of cost-effectiveness data, therefore we should be optimising prescribing of existing drugs to all who can benefit. AChEIs and memantine are very cheap, available in multiple forms, with evidenced benefits to cognition, and their potential benefit to all eligible patients need to be optimised. As newer drugs do become available, there is a risk inequity will worsen [41,42]. Our findings suggest further work is urgently needed to understand primary-care knowledge of memantine prescribing, particularly initiation of co-prescriptions.

Clinicians should be aware of these findings, particularly inequity in prescribing for people from areas of lower SES, ethnic minorities, and with LD. The findings for people with LD require further exploration, examining if these are present

for other guideline-consistent care indicators. The 24% people prescribed memantine is encouraging, but likely far more are eligible; further work is needed to understand barriers to prescribing, and perspectives of GPs about initiating memantine [10], data this resource was unable to capture, particularly given guidance recommending initiating co-prescription in primary care. The decline in prescribing of anti-dementia medications in later years of the study was expected due to study design. This trend needs monitoring, especially as the number of people receiving AChEIs also declined, which is less easily explained given these are indicated in mild AD. Annual reviews and care plans within primary care are a key part of healthcare for people with dementia [4]. These are financially incentivised by the Quality and Outcomes Framework (QOF) [43], and guidance suggests these would include a review of medications, including eligibility for anti-dementia medications [4]. Evidence suggests that these reviews are happening at high rates across subtypes [44], but that quality is variable and patients are not always aware these have taken place [45,46]. Clinicians conducting these annual reviews should be aware of these findings, and for those with AD or LBD dementias ensure that appropriate anti-dementia therapy has been considered.

As the number of people with dementia increases and healthcare evolves and shifts to the community [47,48], we need to ensure we utilise resources we already have equitably, before we distribute new, more expensive, complex drugs for dementia [41,42,49,50].

## Conclusion

This study identified lower than expected rates of prescribing of evidence-based, safe, effective anti-dementia medications. We also identified significant disparities in prescribing based on patient-level characteristics. These disparities were seen in the prescribing of AChEIs with IMD, ethnicity, sex, and LD. For memantine disparities were seen with sex, LD, and ethnicity. These findings require urgent clinical and academic action. As dementia healthcare increasingly shifts to primary care, it is important that existing effective treatments are utilised equitably, laying the groundwork for fair access to future therapies.

## Supporting information

**S1 Fig. Description of the cohort study design.**
(PDF)

**S2 Fig. Criteria for incident co-prescribing (AChE-inhibitor and memantine), Criteria for incident switching (AChE-inhibitor to memantine),Criteria for prevalent switch (AChE-inhibitor to memantine).**
(PDF)

**S1 Table. Number of people ever receiving anti-dementia medications, based on year of diagnosis; Number of people receiving the first prescription of AChEIs and memantine each year, as a percentage of the total number in the study each year.**
(PDF)

**S2 Table. Trends in prescribing of both memantine and AChEIs over the study period; Incidence of switching from AChEIs to memantine in preceding 28 days;Prevalence of switching from AChEIs to memantine.**
(PDF)

**S3 Fig. Sensitivity analysis examining people ever issued acetyl-cholinesterase inhibitors, including people with unspecified dementia, AD, LBD and mixed (AD/LBD) subtypes.** (n = 425,332); Sensitivity analysis examining people ever issued memantine including people with unspecified dementia, AD, LBD and mixed (AD/LBD) subtypes. (n = 425,332).
(PDF)

**S4 Fig. Percentage of patients prescribed acetyl-cholinesterase inhibitors for each practice by patient-level deprivation quintile (violin plots) (n = 1489 practices);Percentage of patients prescribed memantine for each practice by patient-level deprivation quintile (violin plots) (n = 1489 practices).**
(PDF)

**S5 Fig. Competing risks model for ever issued acetyl-cholinesterase inhibitor (n = 24,252) [run on random 10% sample];Competing risks model for ever issued memantine (n = 24,252) [run on random 10% sample].**
(PDF)

**S6 Fig. Parametric regression model: Adjusted hazard ratio for likelihood of ever receiving an acetyl-cholinesterase inhibitor based on patient characteristics (n = 242,007);Parametric regression model: Adjusted hazard ratio for likelihood of ever receiving memantine based on patient characteristics (n = 242,007).**
(PDF)

**S3 Table. Accelerated failure time model with lognormal distribution (acetyl-cholinesterase inhibitor, n = 242007); Accelerated failure time model with lognormal distribution (memantine, n = 242007).**
(PDF)

**S7 Fig. Adjusted hazard ratio for likelihood of ever receiving acetyl-cholinesterase inhibitors based on patient characteristics within 24 months of index date, (Cox regression) (n = 242,007); Adjusted hazard ratio for likelihood of ever receiving memantine based on patient characteristics within 24 months of index date, (Cox regression) (n = 242,007).**
(PDF)

**S4 Table. Coefficients and Time Interaction Coefficients from flexible parametric models (AChEIs) (n = 242,007); Coefficients and Time Interaction Coefficients from flexible parametric models (memantine) (n = 242,007).**
(PDF)

## Acknowledgments

This study is based on data from the Clinical Practice Research Datalink obtained under licence from the UK Medicines and Healthcare Products Regulatory Agency (MHRA). The data are provided by patients and collected by the NHS as part of its care and support. The interpretation and conclusions in this study are those of the authors alone, and not necessarily those of the MHRA, National Institute of Health and Care Research, NHS, or Department of Health and Social Care. The study protocol was approved by Clinical Practice Research Datalink's independent scientific advisory committee (reference 23_002686). We acknowledge all of the data providers and general practices who make anonymised data available for research.

We would also like to thank the PRIMER patient and public participation group, from The University of Manchester for their ongoing input into the study. We would also like to thank the DRAGON patient and public involvement group at Salford University, for their input and ideas around dissemination.

## Author contributions

**Conceptualization:** Charlotte Morris, Louise Robinson, Louise Allan, Darren M. Ashcroft, Tom Blakeman, Evangelos Kontopantelis.

**Data curation:** Charlotte Morris, Pearl L. H. Mok, Darren M. Ashcroft.

**Formal analysis:** Charlotte Morris.

**Funding acquisition:** Charlotte Morris.

**Investigation:** Charlotte Morris.

**Methodology:** Charlotte Morris.

**Project administration:** Charlotte Morris.

**Supervision:** Louise Robinson, Louise Allan, Darren M. Ashcroft, Tom Blakeman, Evangelos Kontopantelis.

**Writing – original draft:** Charlotte Morris.

**Writing – review & editing:** Pearl L. H. Mok, Louise Robinson, Louise Allan, Darren M. Ashcroft, Tom Blakeman, Evangelos Kontopantelis.

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
