## [Decision Letter · Decision Letter 0]

8 Mar 2026

PONE-D-26-03808Prescribing of anti-dementia medications in primary care: a retrospective cohort study in 1489 English General PracticesPLOS One

Dear Dr. Morris,

Thank you for submitting your manuscript to PLOS ONE. It delivers an important message. Both reviewers have been positive but have made some useful suggestions for improvement. After careful consideration, therefore, we feel that your manuscript has merit but does not fully meet PLOS ONE’s publication criteria as it currently stands. We invite you to submit a revised version of the manuscript that addresses the points raised during the review process.

We look forward to receiving your revised manuscript.

Kind regards,

Antony Bayer

Academic Editor

PLOS One

Journal Requirements:

“This study was funded by the National Institute for Health and Care Research (NIHR) School for Primary Care Research [Grant reference: WT6473650](CM). TB, PLHM and DMA are funded by the NIHR Greater Manchester Patient Safety Research Collaboration [Grant NIHR204295]. LA is supported by the Peninsula NIHR Applied Research Collaboration and the Exeter NIHR Biomedical Research Centre. The views expressed in this publication are those of the authors and not necessarily those of the NHS, the National Institute for Health and Care Research, or the Department of Health and Social Care.”

6. Please upload a copy of Figures 1 and 2, to which you refer in your text on page 5 and 8. If the figure is no longer to be included as part of the submission please remove all reference to it within the text.

Reviewers' comments:

Reviewer's Responses to Questions

Comments to the Author

1. Is the manuscript technically sound, and do the data support the conclusions?

Reviewer #1: Partly

Reviewer #2: Yes

2. Has the statistical analysis been performed appropriately and rigorously? 

Reviewer #1: Yes

Reviewer #2: Yes

3. Have the authors made all data underlying the findings in their manuscript fully available?

Reviewer #1: Yes

Reviewer #2: Yes

4. Is the manuscript presented in an intelligible fashion and written in standard English?

Reviewer #1: Yes

Reviewer #2: Yes

5. Review Comments to the Author

Reviewer #1: This paper addresses some important questions about the prescribing of drugs for dementia in Primary Care and the impact of socioeconomic status and ethnicity. However, I have the following comments:

Abstract Objective: does not mention cholinesterase inhibitors (AChEIs) specifically only memantine. As these drugs are a major interest of the study this should be corrected

Dementia Severity is a key difference in determining whether a person is started on treatment with an AChEI or memantine. The fact that the study was unable account for the effect of this on prescribing is a major limitation of the study. This is only mentioned in the discussion. This is an important issue that should receive more discussion and mention in methodology.

Tolerability of AChEIs: Memantine is indicated for people who are unable to tolerate AChEIs, and monotherapy with memantine would be appropriate for such patients; again it is not mentioned whether of not this could be determined from the CPRD database. This point is not considered in the discussion.

Coding Issues: apart from limitations of the CPRD database other coding issues are not discussed. It is well known that ethnicity is poorly coded in Primary care. This issue, how missing data was handled and the impact on the analysis and findings should be discussed.

Other Factors: Two other things happened between 2006 and 2024

The DiADeM (Diagnosing Advanced Dementia Mandate) project, designed for use in care homes in England, was launched as a national pilot in 14 sites by NHS England in 2022. Some of these sites were included in the areas included in this study. Memantine would be more likely to be used in people with advanced dementia. The implications of this need to be considered as a reason for monotherapy prescriptions in the data

COVID: the impact of Covid delaying diagnosis and on initiating prescribing of AChEIs and / or memantine should be discussed. During Covid there was an upsurge in the prescription of antipsychotic drugs for behavioural issues – were these used instead of trial of memantine?

Reviewer #2: This is. very important paper that demonstrates a lack of prescribing and inequality of prescribing effective, safe & cheap medications for several types of dementia.

I would suggest the following edits

In the introduction I suggest adding this meta analysis as the efficacy of ACEI is higher in DLB than AD and this may not be well known and may reduce prescribing in that condition

Knight R, Khondoker M, Magill N, Stewart R, Landau S. A Systematic Review and Meta-Analysis of the Effectiveness of Acetylcholinesterase Inhibitors and Memantine in Treating the Cognitive Symptoms of Dementia. Dement Geriatr Cogn Disord. 2018;45(3-4):131-151. doi: 10.1159/000486546. Epub 2018 May 7. PMID: 29734182.

This may explain why prescribing for ACHEI is higher in DLB. This is good but still a large percentage of eligible patients are not being prescribed

In the section ‘Implications for practice policy and research ‘

There is a sentence, ‘New curative drugs for AD33, 34 are currently unavailable, due to a lack of cost-effectiveness data, therefore we should be optimising prescribing of existing drugs to all who can benefit. ‘

Please change to

New disease modifying anti-amyloid drugs for AD33, 34 are currently unavailable in the NHS, due to a lack of cost-effectiveness data, therefore we should be optimising prescribing of existing drugs to all who can benefit. ‘

They do not cure AD. There is no cure. The drugs are available, but you have to pay for them. Approximate cost is £70K to include prescribing drugs and MRI monitoring

Could the authors comment or add in that these drugs are very cheap (see below)

COST of Donepezil

• Standard Tablet Cost: According to the NHS Drug Tariff (Part VIIIA Category M), the basic price for 28 tablets of donepezil (both 5mg and 10mg) is often in the region of £0.94 to £1.04.

• Oral Solution: 150ml of 1mg/ml sugar-free oral solution is priced at approximately £118.63.

Based on the NHS Electronic Drug Tariff and formulary, the costs for rivastigmine vary by formulation (capsules or patches) and strength. As of recent data, generic rivastigmine is generally cost-effective, but specific formulations can differ:

• Capsules (28-pack, various strengths): Prices generally range from £1.70 to £2.40 for 28 capsules, depending on the manufacturer.

• Transdermal Patches (13.3mg/24hours, 30 pack): The basic price is approximately £77.97.

• Oral Solution (2mg/ml, 120ml): The basic price is approximately £96.82 - £99.14.

• Orodispersible Tablets: These are very expensive, with 10mg tablets costing roughly 5 times more than 5mg tablets.

Based on the NHS Drug Tariff and BNF (British National Formulary), the cost of memantine varies depending on the form, strength, and whether the branded version (Ebixa) or generic is prescribed. Generic memantine is significantly cheaper for the NHS.

Approximate NHS Costs (Drug Tariff Part VIIIA, 2026):

• Memantine 10mg Tablets (Pack of 28): Roughly £1.24.

• Memantine 20mg Tablets (Pack of 28): Roughly £1.91.

• Memantine 10mg/ml Oral Solution (50ml): Approximately £8.94.

• Orodispersible Tablets (Sugar-free, 28 tablets): These are much more expensive, often costing between £12.50 (5mg) and £49.98 (20mg)

Could the authors add anything about safety. Can they see how many people were prescribed these drugs and could not tolerate them?

In the implication section can the authors include that Annual reviews by GPs are a key part of a person with dementia's care.

Annual dementia reviews are funded by the NHS as part of primary care (general practice) services, specifically under the Quality and Outcomes Framework (QOF) in England. They do not need to be undertaken by a GP but should bestructured annual review with a GP or nurse to monitor symptoms, medication, and support needs

Could the implication comment on are these annual reviews monitored. Could medication revie be included. Could pharmacists flag up cases where there is a demenatoi diagnosis and where there are no ACHEI or memantine prescribed?

6. PLOS authors have the option to publish the peer review history of their article (what does this mean?). If published, this will include your full peer review and any attached files.

Do you want your identity to be public for this peer review? For information about this choice, including consent withdrawal, please see our Privacy Policy.

Reviewer #1: No

Reviewer #2: No

---

## [Author Response · Author response to Decision Letter 1]

27 Mar 2026

Dear Editor and Reviewers,

Thank you for taking the time to review our manuscript and provide the detailed feedback on the paper. The comments you have provided are helpful and strengthen the manuscript. Please see our detailed responses to your comments below. We hope these address your concerns regarding the paper.

---

1. Is the manuscript technically sound, and do the data support the conclusions?

Reviewer #1: Partly

Reviewer #2: Yes

Thank you for the comments. After the suggested revisions, we hope any concerns regarding the rigour of the research are addressed for reviewer 1.

2. Has the statistical analysis been performed appropriately and rigorously?

Reviewer #1: Yes

Reviewer #2: Yes

Thank you for the comments

3. Have the authors made all data underlying the findings in their manuscript fully available?

The PLOS Data policy [track.editorialmanager.com] requires authors to make all data underlying the findings described in their manuscript fully available without restriction, with rare exception (please refer to the Data Availability Statement in the manuscript PDF file). The data should be provided as part of the manuscript or its supporting information, or deposited to a public repository. For example, in addition to summary statistics, the data points behind means, medians and variance measures should be available. If there are restrictions on publicly sharing data—e.g. participant privacy or use of data from a third party—those must be specified.

Reviewer #1: Yes

Reviewer #2: Yes

Thank you for the comments

4. Is the manuscript presented in an intelligible fashion and written in standard English?

Reviewer #1: Yes

Reviewer #2: Yes

Thank you for the comments.

5. Review Comments to the Author

Reviewer #1: This paper addresses some important questions about the prescribing of drugs for dementia in Primary Care and the impact of socioeconomic status and ethnicity.

Thank you for agreeing about the importance of the work.

However, I have the following comments:

Abstract Objective: does not mention cholinesterase inhibitors (AChEIs) specifically only memantine. As these drugs are a major interest of the study this should be corrected

Thank you for pointing this out. It is important the abstract is clear, and it has been amended as suggested:

[Sentence added to: Abstract, OBJECTIVE]

Patterns of prescribing of Acetyl Cholinesterase Inhibitors (AChEIs) and memantine require further exploration

Dementia Severity is a key difference in determining whether a person is started on treatment with an AChEI or memantine. The fact that the study was unable account for the effect of this on prescribing is a major limitation of the study. This is only mentioned in the discussion. This is an important issue that should receive more discussion and mention in methodology.

We thank the reviewer for this insightful point. We agree that dementia severity is an important factor in prescribing decisions. We have expanded our discussion of this limitation and added clarifying text to the methods section to explain our approach.

In primary care electronic health records, the coding of even a dementia diagnosis is frequently suboptimal or absent1-3, although evidence is limited, coding of severity of dementia is likely even less accurate and more variable4. Restricting our cohort only to patients with a documented severity level at baseline would introduce significant selection bias and drastically reduce our sample size, potentially undermining the study’s power.

As dementia is a progressive condition, a mild baseline diagnosis often transitions to moderate or severe during the follow-up period. Furthermore, clinical guidelines evolved during our study period (as shown in Figure 1). This would complicate any analyses based on severity.

Under current and historical guidance (figure 1), the vast majority of patients with AD or LBD are eligible for either an AChEI or memantine at some stage of their illness. By including all patients regardless of documented severity, we capture the real-world clinical trajectory more holistically than a restricted documented severity subgroup would allow.

Sentences have been added in regarding this:

[Method: Classifying Dementia Subtype: Assessing Dementia Severity]

Assessing Dementia Severity

We were unable to account for dementia severity at diagnosis due to the complexity of coding patterns for dementia within CPRD. There are codes such as ‘advanced dementia’ or ‘mild dementia’, however coding practices for even a diagnosis of dementia are variable and suboptimal1-3. Evidence is limited, but the accuracy and frequency of severity data is likely even worse4. Severity may also be recorded as free text which we were unable to access for analyses. Furthermore, any codes detailing severity may account for severity at diagnosis, but not how the condition progresses so if a mild dementia goes on to become moderate (as is clinically expected), the code may not change.

[Discussion: Strengths and Limitations]

This meant we could not determine eligibility for medications at baseline, or whether AChEIs or memantine should be offered. However, because dementia is a progressive condition a single baseline severity measure may not reflect eligibility for treatment over the entire study period. In clinical reality, most people with a diagnosis would be eligible for at least one of the examined medications at some point during the clinical trajectory (AChEI if diagnosed with mild or moderate; memantine if diagnosed when severe). This is an important limitation, and highlights the need for more detailed standardised dementia coding within primary care.

Tolerability of AChEIs: Memantine is indicated for people who are unable to tolerate AChEIs, and monotherapy with memantine would be appropriate for such patients; again it is not mentioned whether of not this could be determined from the CPRD database. This point is not considered in the discussion.

Thank you for this helpful comment. You are entirely correct here with this point. Although codes suggesting medications were not tolerated are available, these would be inconsistently applied and limit the sample considerably and would introduce bias into analyses. Therefore, while we agree this is an important limitation, we also believe our methodology examining issues of prescriptions is reasonable and robust enough to answer our research questions. Nevertheless, a sentence recognising this important limitation has been added.

[Discussion: Strengths and Limitations].

It was not possible to assess if patients were offered AChEIs or memantine and declined or did not tolerate the medications.

Coding Issues: apart from limitations of the CPRD database other coding issues are not discussed. It is well known that ethnicity is poorly coded in Primary care. This issue, how missing data was handled and the impact on the analysis and findings should be discussed.

Thank you for the important point. Further sentences and important references have been added to the text to address this. By using the CPRD linked ethnicity data, we have used the best available data for ethnicity5. It was derived using CPRD’s algorithm which uses not only primary but also secondary care level data to provide an ethnicity value for each participant in the study. Previously, data for ethnicity were incomplete and poorly coded within primary care. However, the data for ethnicity recording within CPRD have improved in recent years and area not of high quality5. Missingness for ethnicity or IMD at the patient-level was very small for eligible patients (n=249, <0.04%); complete case analysis was conducted and given the small number of missing cases it is unlikely to have influenced results.

For other conditions, again missingness was low, because we deemed a code being absent as the condition not being present, not missing. We could have explored this further by looking at proxies for important diagnoses, e.g. issues of medications specific to each of the comorbidities included to increase accuracy. This was a limitation, which we have now discussed in the limitations section. However, as the comorbidity score was used as a covariate to adjust for chronic conditions, use of a code for the condition rather than detailed case finding is reasonable. We had wide ranging codelists, and considered a single instance of a code as the patient having the condition. All of the 19 comorbidities included are important medical conditions; if someone has one of these it is reasonable to assume that at some point a single code will have been entered into their primary care record.

[Methods: Demographics and comorbidities]

If a condition was not coded, then it was deemed absent not missing.

These represent the best available data for ethnicity within CPRD, and evidence suggests that CPRD ethnicity and IMD data is representative of the overall population and suitable for use in research5, 6

[Discussion: Strengths and Limitations]

We took absence of a diagnostic code as absence of a condition in terms of the comorbidity score. Although this means we may have underestimated the comorbidity burden for some patients, as we had very wide ranging codelists and we used a single instance of the code as identifying the comorbidity, the number of patients affected is likely to be minimal

Other Factors: Two other things happened between 2006 and 2024: The DiADeM (Diagnosing Advanced Dementia Mandate) project, designed for use in care homes in England, was launched as a national pilot in 14 sites by NHS England in 2022. Some of these sites were included in the areas included in this study. Memantine would be more likely to be used in people with advanced dementia. The implications of this need to be considered as a reason for monotherapy prescriptions in the data

COVID: the impact of Covid delaying diagnosis and on initiating prescribing of AChEIs and / or memantine should be discussed. During Covid there was an upsurge in the prescription of antipsychotic drugs for behavioural issues – were these used instead of trial of memantine?

Thank you for raising these important points. We have addressed them by referencing the time series analyses which did not show dramatic shifts around the time of the covid pandemic, or when drives for diagnosis were being conducted by NHS England.

[Discussion: Results in the context of other literature]

The study covers the period of the Covid-19 pandemic (2020-22), however the time series analyses did not show any dramatic changes in prescribing levels over these months. This may be because prescribing continued, or because there were fewer diagnoses and fewer prescriptions issued, meaning overall percentages remained the same. Other large important studies going on during this period included NHS England drives to diagnose dementia7, 8 and changes to guidance as shown in figure 1Despite thiswe did not see notable shifts in monthly prescribing levels that corresponded with these events.

---

Reviewer #2: This is. very important paper that demonstrates a lack of prescribing and inequality of prescribing effective, safe & cheap medications for several types of dementia.

Thank you for the comment about the importance of the study, with which we fully agree.

I would suggest the following edits

In the introduction I suggest adding this meta analysis as the efficacy of ACEI is higher in DLB than AD and this may not be well known and may reduce prescribing in that condition

Knight R, Khondoker M, Magill N, Stewart R, Landau S. A Systematic Review and Meta-Analysis of the Effectiveness of Acetylcholinesterase Inhibitors and Memantine in Treating the Cognitive Symptoms of Dementia. Dement Geriatr Cogn Disord. 2018;45(3-4):131-151. doi: 10.1159/000486546. Epub 2018 May 7. PMID: 29734182.

This may explain why prescribing for ACHEI is higher in DLB. This is good but still a large percentage of eligible patients are not being prescribed

Thank you for pointing out this important and interesting reference. It is highly relevant and also important to highlight this to readers to increase awareness. A sentence has been added and the reference.

[Introduction]

Systematic-review evidence suggests that the efficacy of AChEIs is actually higher in those with LBD than AD9, but this may not be well known particularly among non-specialists within primary care, and lead to lower rates of prescribing in this subtype.

In the section ‘Implications for practice policy and research ‘

There is a sentence, ‘New curative drugs for AD33, 34 are currently unavailable, due to a lack of cost-effectiveness data, therefore we should be optimising prescribing of existing drugs to all who can benefit. ‘

Please change to

New disease modifying anti-amyloid drugs for AD33, 34 are currently unavailable in the NHS, due to a lack of cost-effectiveness data, therefore we should be optimising prescribing of existing drugs to all who can benefit. ‘

They do not cure AD. There is no cure. The drugs are available, but you have to pay for them. Approximate cost is £70K to include prescribing drugs and MRI monitoring

Thank you for this detailed comment and your attention to detail here. The sentence has been changed as suggested.

[Implications for practice, policy and research]

New disease modifying anti-amyloid drugs10, 11 are currently unavailable on the NHS, due to a lack of cost-effectiveness data, therefore we should be optimising prescribing of existing drugs to all who can benefit

Could the authors comment or add in that these drugs are very cheap (see below)

COST of Donepezil

• Standard Tablet Cost: According to the NHS Drug Tariff (Part VIIIA Category M), the basic price for 28 tablets of donepezil (both 5mg and 10mg) is often in the region of £0.94 to £1.04.

• Oral Solution: 150ml of 1mg/ml sugar-free oral solution is priced at approximately £118.63.

Based on the NHS Electronic Drug Tariff and formulary, the costs for rivastigmine vary by formulation (capsules or patches) and strength. As of recent data, generic rivastigmine is generally cost-effective, but specific formulations can differ:

• Capsules (28-pack, various strengths): Prices generally range from £1.70 to £2.40 for 28 capsules, depending on the manufacturer.

• Transdermal Patches (13.3mg/24hours, 30 pack): The basic price is approximately £77.97.

• Oral Solution (2mg/ml, 120ml): The basic price is approximately £96.82 - £99.14.

• Orodispersible Tablets: These are very expensive, with 10mg tablets costing roughly 5 times more than 5mg tablets.

Based on the NHS Drug Tariff and BNF (British National Formulary), the cost of memantine varies depending on the form, strength, and whether the branded version (Ebixa) or generic is prescribed. Generic memantine is significantly cheaper for the NHS.

Approximate NHS Costs (Drug Tariff Part VIIIA, 2026):

• Memantine 10mg Tablets (Pack of 28): Roughly £1.24.

• Memantine 20mg Tablets (Pack of 28): Roughly £1.91.

• Memantine 10mg/ml Oral Solution (50ml): Approximately £8.94.

• Orodispersible Tablets (Sugar-free, 28 tablets): These are much more expensive, often costing between £12.50 (5mg) and £49.98 (20mg)

Thank you for the detailed comment, with whic

---

## [Editor Report · Decision Letter 1]

9 Apr 2026

Prescribing of anti-dementia medications in primary care: a retrospective cohort study in 1489 English General Practices

PONE-D-26-03808R1

Dear Dr. Morris,

Thank you for your revised manuscript and for your careful attention to addressing the reviewers' comments. We’re pleased to inform you that your manuscript has been judged scientifically suitable for publication and will be formally accepted for publication once it meets all outstanding technical requirements.

Kind regards,

Antony Bayer

Academic Editor

PLOS One
---

## [Editor Report · Acceptance letter]

PONE-D-26-03808R1

PLOS One

Dear Dr. Morris,

I'm pleased to inform you that your manuscript has been deemed suitable for publication in PLOS One. Congratulations! Your manuscript is now being handed over to our production team.

Kind regards,

on behalf of

Professor Antony Bayer

Academic Editor

PLOS One